# Achieving ultrahigh triboelectric charge density for efficient energy harvesting

Jie Wang[1,2], Changsheng Wu[2], Yejing Dai[2,3], Zhihao Zhao[3], Aurelia Wang[2], Tiejun Zhang[2] & Zhong Lin Wang[1,2]

With its light weight, low cost and high efficiency even at low operation frequency, the triboelectric nanogenerator is considered a potential solution for self-powered sensor networks and large-scale renewable blue energy. As an energy harvester, its output power density and efficiency are dictated by the triboelectric charge density. Here we report a method for increasing the triboelectric charge density by coupling surface polarization from triboelectrification and hysteretic dielectric polarization from ferroelectric material in vacuum ($P \sim 10^{-6}$ torr). Without the constraint of air breakdown, a triboelectric charge density of $1003\,\mu C\,m^{-2}$, which is close to the limit of dielectric breakdown, is attained. Our findings establish an optimization methodology for triboelectric nanogenerators and enable their more promising usage in applications ranging from powering electronic devices to harvesting large-scale blue energy.

[1] Beijing Institute of Nanoenergy and Nanosystems, Chinese Academy of Sciences, National Center for Nanoscience and Technology (NCNST), Beijing 100083, China. [2] School of Materials Science and Engineering, Georgia Institute of Technology, Atlanta, GA 30332, USA. [3] Key Laboratory for Advanced Ceramics and Machining Technology, Ministry of Education, School of Material Science and Engineering, Tianjin University, Tianjin 300072, China. Jie Wang, Changsheng Wu and Yejing Dai contributed equally to this work. Correspondence and requests for materials should be addressed to Z.L.W. (email: zhong.wang@mse.gatech.edu)

ntensive research efforts have been devoted to sustaining the huge energy consumption of modern society while minimizing the environmental cost. Harvesting energy from renewable natural resources such as the sun, the wind and biomass, has been demonstrated to be a sustainable alternative for energy crisis, and plays an increasingly important role with the fast depletion of fossil fuels[1]. With their light weight, low cost and high efficiency even at low operation frequency, the newly invented triboelectric nanogenerators (TENGs) have been proven to be a promising complementary solution for harvesting ambient mechanical energy that is ubiquitous but wasted in our everyday life[2–8]. The operation of TENGs is based on triboelectrification (or contact electrification) and electrostatic induction[9], and the fundamental theory lies in Maxwell's displacement current and change in surface polarization[10]. Previous work on TENG has demonstrated its potential of wide application ranging from powering small electronic devices for self-powered systems, to functioning as active sensors for medical, infrastructural, human–machine, environmental monitoring, and security[11–17]. Moreover, it can be effectively used for scavenging energy from low frequent ocean waves for the prospect of large-scale blue energy[18–21].

As an energy harvester, the commercialization and application of TENGs highly depend on their power density, which is quadratically related to the triboelectric charge density[22]. Therefore, large efforts have been devoted to increasing the amount of triboelectric charges by means of material improvement, structural optimization, surface modification, and so on[23–26]. Artificial injection of ions, for example, by using corona discharging, was considered a straight forward way to increase the charge density,

resulting in a high output charge density of 240 μC m$^{-2}$, but long-term stability remains an issue[27]. Very recently, a high-output charge density of 250 μC m$^{-2}$ was realized on a TENG through elastomeric materials and a fragmental contact structure[28]. Nevertheless, the achievable output charge density has still limited been by the phenomenon of air breakdown in all previous studies.

For triboelectrification in the air atmosphere, it is also known that the effective contact area is significantly smaller than the overall surface area, and thus optimizing the contact area and structure of the TENG can effectively increase the overall triboelectric charge density. Herein, we first demonstrate that with the improved soft-contact and fragmental structure, the triboelectric charge density can be increased from 50 to 120 μC m$^{-2}$ in air when compared to a conventional TENG with only hard contact. By applying high vacuum (~10$^{-6}$ torr), the charge density is further boosted to 660 μC m$^{-2}$ without the limitation of air breakdown. With the coupling of surface polarization from triboelectrification and hysteretic dielectric polarization from a ferroelectric material, it can jump to 1003 μC m$^{-2}$, which elevates the maximum output power density of a conventional TENG from 0.75 to 50 W m$^{-2}$ even at a low-motion frequency of about 2 Hz, the normal frequency of human walking and ocean waves. These findings open more possibilities for TENGs both as highly efficient mechanical energy harvesters for large-scale energy sources such ocean waves, and as self-powered modules integratable with devices beyond wearable electronics and sensors. Our findings may also give new insights into long-lasting debates over the mechanism of triboelectrification and its kinetics.

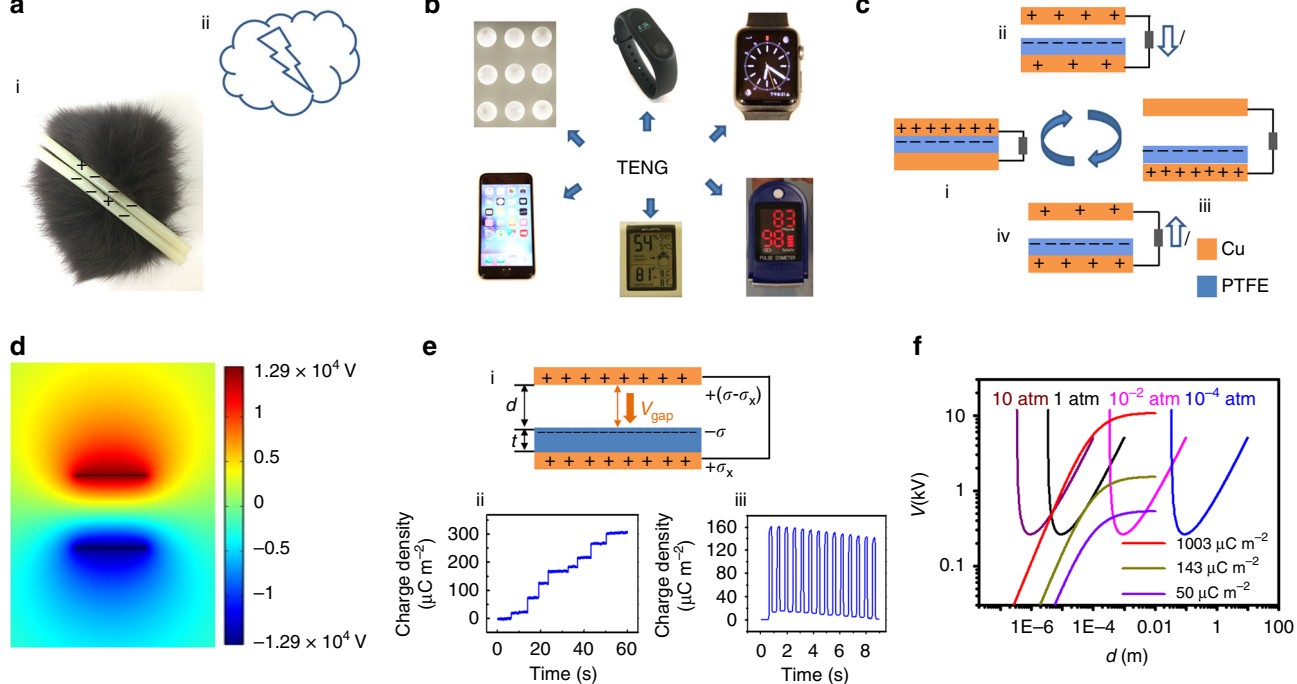

**Fig. 1** Contact electrification and air breakdown. **a** Phenomena of contact electrification in nature, such as lighting and triboelectrification of rubber rod and fur. **b** Applications of triboelectric nanogenerators (TENGs) for driving electronic devices, such as electric bulb, fitness tracker, electric watch, health monitor, temperature-humidity meter, and cell phones. **c** Working mechanism of contact-separation TENG. **d** Finite-element simulation of the potential distribution between two triboelectric materials, copper and polytetrafluoroethylene (PTFE) with a gap distance of 1 cm and a surface charge density of 50 μC m$^{-2}$ using COMSOL. **e** Output charge density of TENG after ion air injection. *Top insert* shows schematic of gap voltage ($V_{gap}$) between the metal electrode and the dielectric film, which could cause the breakdown of the air; *bottom-left insert* records the in situ measurement of the charge flow from the ground to the bottom electrode of the PTFE film during the step-by-step ion injection process; *bottom-right insert* plots the charge transfer between the two electrodes of TENG operating after the ion injection. **f** Air breakdown voltage at different pressures and gap voltage of TENG with different charge densities. The thickness of PTFE film is 200 μm

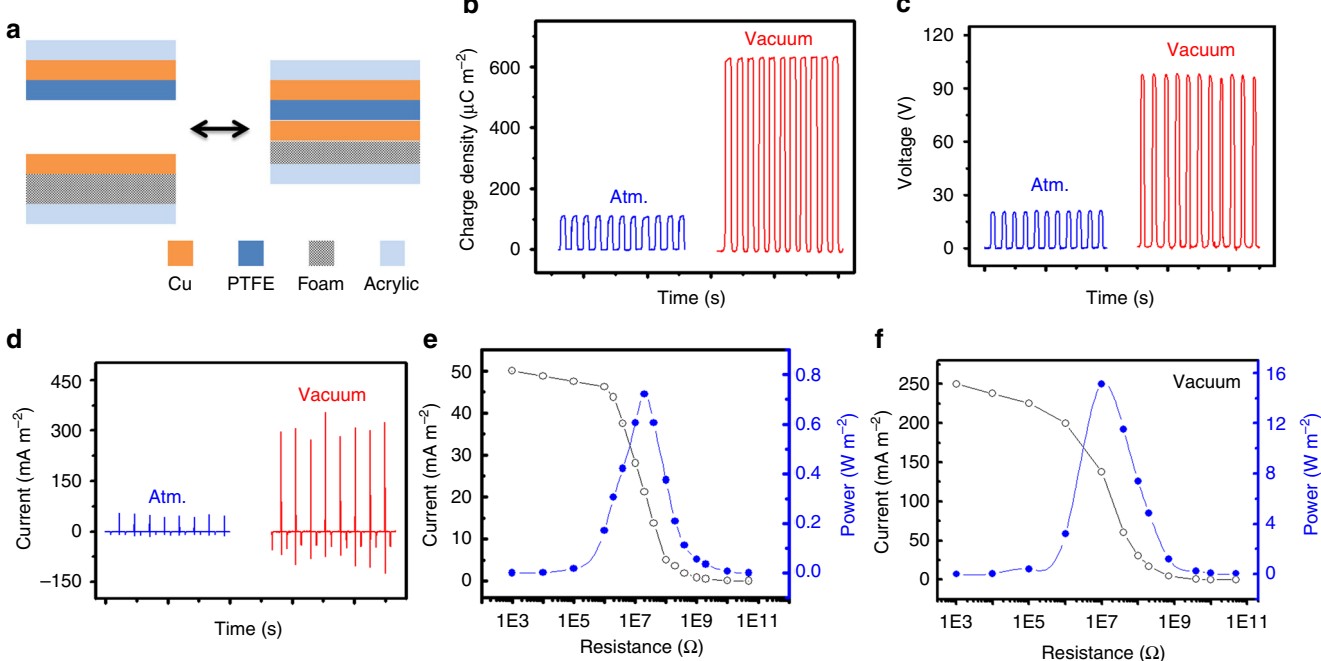

**Fig. 2** Output performance of TENG in the vacuum. **a** Schematic of TENG with a cushioned Cu electrode to increase contact intimacy during operation process. **b**–**d**,**b** Charge density, **c** open-circuit voltage, and **d** short-circuit current of the TENG in atmosphere and high vacuum ($P \sim 10^{-6}$ torr). **e**,**f** Current density and power density of TENG with various loads in atmosphere **e** and vacuum **f**

## Results

**Contact electrification and air breakdown.** Contact electrification, or triboelectrification, refers to the charge transfer between two surfaces in contact, and is one of the oldest and most ubiquitous phenomena in nature, including lighting and amber effect (Fig. 1a)[29]. It is generally considered a negative effect in the electrical and electronic industry, and often causes damages and safety hazards such as shocks and explosions to electronic devices and equipment. Its positive applications have been limited to areas such as laser printing, photocopying, and electrostatic separations.

In 2012, this effect was first applied to develop the new energy harvesting technology—TENG[2]. By coupling the triboelectric effect and electrostatic induction, a TENG can generate alternating-current electrical outputs, and is capable of driving many electronic devices and even charging mobile phones (Fig. 1b)[30–32]. Typically, a simple TENG consists of two metal electrodes and a dielectric film, for example, a top copper (Cu) film and a bottom polytetrafluoroethylene (PTFE) film with a back electrode attached to it (Fig. 1c). When the two films are in contact, the Cu film and the PTFE film acquire net positive and net negative triboelectric charges, respectively. When separated, the resulting charge separation will induce a potential difference across the two Cu electrodes and subsequent current flow if the device is connected to an external circuit. The current flow is reversed when the two charged surfaces are brought into contact again[33].

In particular, a very high electrostatic field can be built between the two separated surfaces with opposite triboelectric charges, e.g., the top Cu film and the bottom PTFE film. The potential distribution between such two films can be numerically simulated using the finite-element method via the commercial software COMSOL. With a gap distance of 1 cm between the Cu and PTFE film and a surface charge density of $50 \mu C\,m^{-2}$, as commonly observed in TENGs, a high gap voltage of $2.6 \times 10^4$ V will occur (Fig. 1d). When ionized-air injection is applied, a charge density of $300 \mu C\,m^{-2}$ can be easily achieved (Fig. 1e) and will induce a

voltage of $1.5 \times 10^5$ V with all other parameters kept the same. Even though these high potentials ($V_{gap}$) are theoretical values under open-circuit conditions, they give a rough idea of the high likelihood of air breakdown when two surfaces with large density of opposite charges are close to each other. The possible breakdown greatly limits the maximum retainable charge density in TENGs as well as other triboelectrification applications. As shown in Fig. 1e, even though an initial high charge density of $300 \mu C\,m^{-2}$ can be obtained, the maximum achieved output charge density under short-circuit conditions is limited to $160 \mu C\,m^{-2}$ by air breakdown and decreases to $140 \mu C\,m^{-2}$ within 8 s.

Paschen's law describes the empirical relationship between the gaseous breakdown voltage ($V_b$) and the product of the gas pressure ($P$) and gap distance ($d$), and is given by

$$V_b = \frac{APd}{\ln(Pd) + B},$$ (1)

where A and B are the constants determined by the composition and the pressure of the gas. For air at standard atmospheric pressure (atm, i.e., the conventional operation condition of a TENG), $A = 2.87 \times 10^5$ V atm$^{-1}$ m$^{-1}$, and $B = 12.6$.

According to the theoretical derivation[22], the gap voltage between contact surfaces of a Cu-PTFE TENG ($V_{gap}$) under short-circuit condition is given by

$$V_{gap} = \frac{t\sigma d}{\varepsilon_0(t + d\varepsilon_r)},$$ (2)

where $t$ is the thickness of the PTFE film, $\sigma$ the triboelectric surface charge density, $\varepsilon_r$ the relative permittivity of PTFE ($\varepsilon_r \sim 2.1$), and $\varepsilon_0$ the vacuum permittivity ($\varepsilon_0 \sim 8.85 \times 10^{-12}$ F m$^{-1}$).

Figure 1f plots the $V_b$ of air at different pressures and $V_{gap}$ of TENG with different charge densities. To avoid air breakdown, the $V_{gap}$ must be smaller than $V_b$ at any operation gap distance ($0 < d < d_{max}$). A simple glance at the curves and equations reveals that either increasing the gaseous pressure (left-shifting

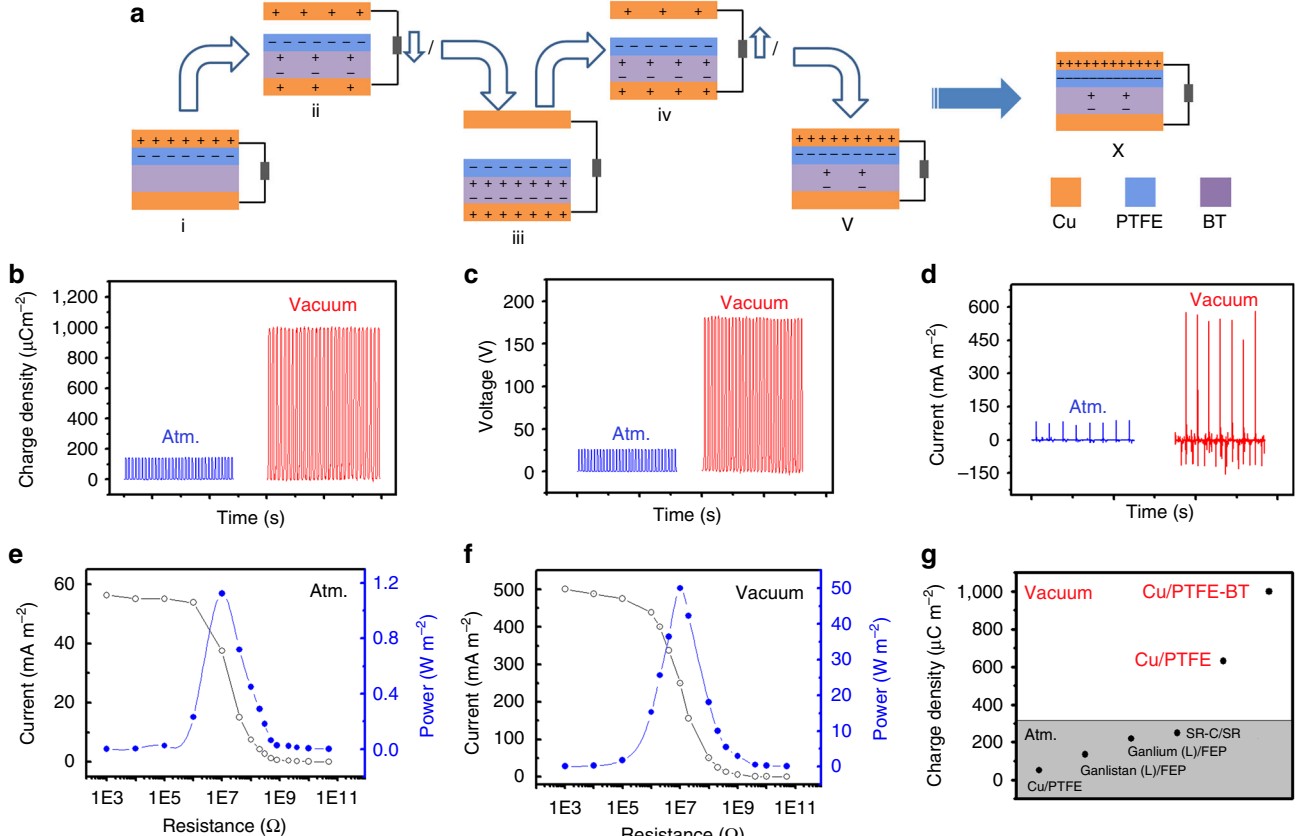

**Fig. 3** Working mechanism and output performance of TENG with the coupling of surface and dielectric polarization. **a** Working mechanism of the TENG with the integration of triboelectric material PTFE with ferroelectric material BT. **b** Charge density, **c** open-circuit voltage, and **d** short-circuit current of the TENG in atmosphere and high vacuum. **e,f** Current density and power density of TENG with various loads at **e** atmosphere and **f** vacuum ($P \sim 10^{-6}$ torr). **g** Comparison of the output charges density measured in this work with previously reported ones

the Paschen Curve) or changing to a gas with higher dielectric strength (up-shifting the Paschen Curve) will allow larger gap voltages without breakdown. However, such improvement is limited and may induce severe safety concerns, such as explosions and leakage of harmful gases. A more effective way is to shift the Paschen's curve toward the right by decreasing the air pressure. Initially, as the pressure decreases, the maximum charge density allowed without breakdown would decrease; once the distance at which the minimum breakdown voltage occurs passes the maximum operation range of TENG, the maximum charge density allowed would increase. In particular, when the pressure is so low that the whole operation range is outside of the region where the Paschen's law holds valid, there would be no concern about avalanche breakdown that would limit the retainable charge density. Therefore, the performance of TENGs under high vacuum was studied and the results are presented in the following sections. The maximum charge density allowed without break down at various air pressures for the TENG with 1 cm operation range is plotted in Supplementary Fig. 1 and Supplementary Note 1. It is worth while to note that at 1 atm, the maximum charge density is 143 µC m$^{-2}$, which corresponds very well with the results shown in Fig. 1e.

**Output performance of TENG in vacuum.** According to our previous report, soft contact and fragmental structure can enhance the output performance of TENG. Hence a TENG with a cushioned Cu electrode and a contact area smaller than a conventional TENG was fabricated as the starting point of our study, with its schematic illustrated in Fig. 2a. While in contact,

the foam conforms to the mechanical stress and thus improves the contact intimacy between the Cu electrode and PTFE. As a result, the triboelectric charge density of the TENG in air measures up to 120 µC m$^{-2}$ (Fig. 2b), about 2.4-fold of that of a conventional TENG without soft contact or fragmental structure (Supplementary Fig. 2 and Supplementary Note 2).

When the same cushioned TENG is operated in vacuum ($P \sim 10^{-6}$ torr), the air breakdown depicted by Paschen's law is avoided and the triboelectric charge density is boosted from 120 to 660 µC m$^{-2}$ (Fig. 2b). Accordingly, the open circuit voltage increases from 20 V to 100 V (Fig. 2c), the peak of short-circuit current rises from 60 to 300 mA m$^{-2}$ (Fig. 2d), and the maximum output power density is enhanced from 0.75 W m$^{-2}$ (Fig. 2e) to 16 W m$^{-2}$ (Fig. 2f) with the match-load of 20 MΩ at a fairly low frequency of 2 Hz.

The effect of the thickness of the dielectric film on the triboelectric charge density was investigated as well. When the thickness of PTFE increases from 200 to 600 µm in air, the charge density decreases from 120 to 90 µC m$^{-2}$ (Supplementary Fig. 3a and Supplementary Note 3). This is expected since the gap voltage increases with the film thickness and it is easier, i.e., possessing lower charge density in this case, to have an air breakdown with a larger gap voltage. As for in vacuum where there is no concern about air breakdown, the triboelectric charge density of TENG with thicker PTFE actually reaches a slightly higher value of 680 µC m$^{-2}$ due to greater contact intimacy, and the output power density is boosted from 0.4 W m$^{-2}$ in air to 20 W m$^{-2}$ (Supplementary Fig. 3b,c), giving a 49-fold enhancement. Therefore, the performance enhancement of TENGs using high vacuum is not limited by the thickness of the dielectric film.

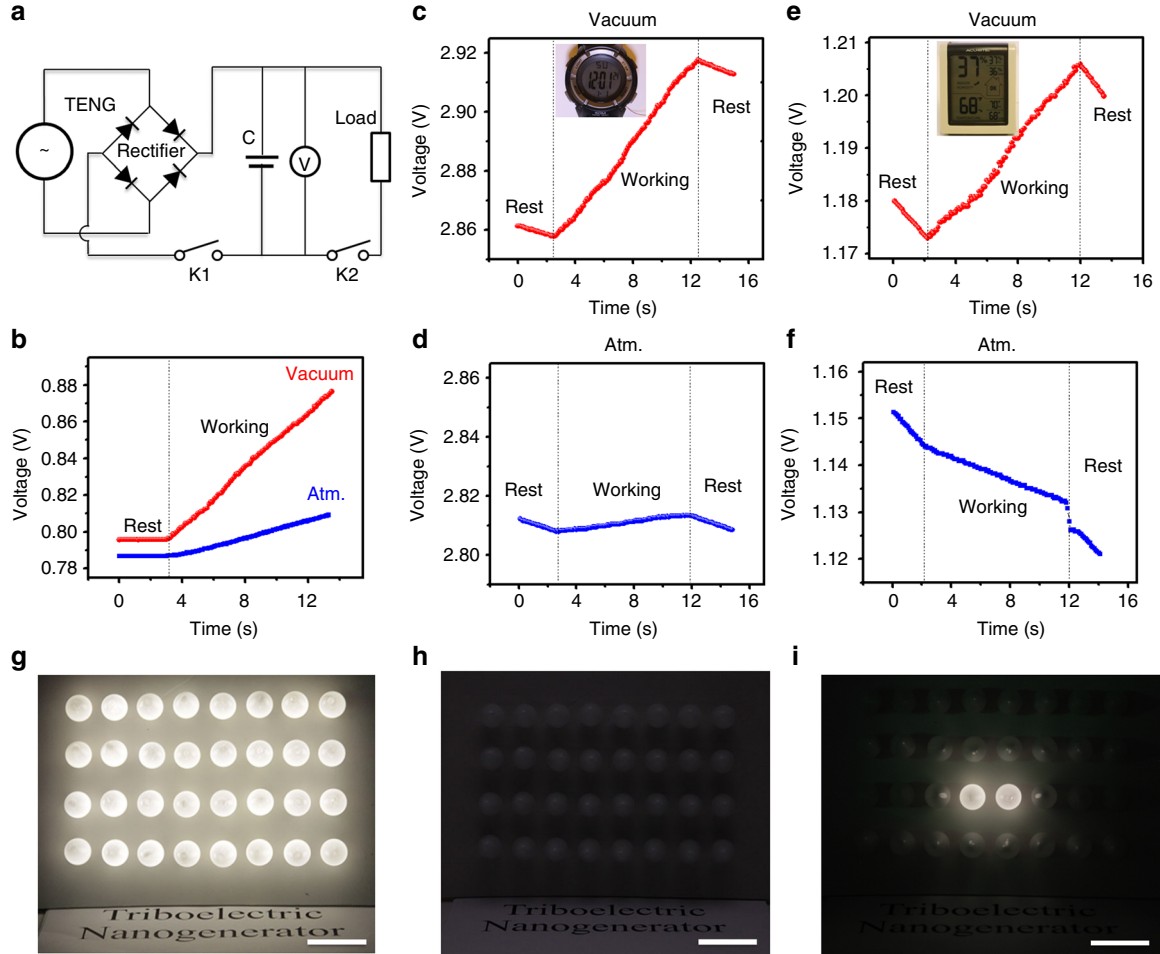

**Fig. 4** Application of the TENG to drive electronics devices. **a** Circuit diagram of the self-powered system consisting of the TENG and a supercapacitor. **b** Charging curves of the supercapacitor by pressing the TENG inatmosphere and high vacuum ($P \sim 10^{-6}$ torr). **c,d** Charging curves of the supercapacitor in high vacuum **c** and in atmosphere **d** when an electronic watch is driven by the TENG simultaneously. **e** Charging curve of the supercapacitor when a temperature–humidity meter is driven by the TENG in high vacuum simultaneously. **f** Discharging curve of the supercapacitor when the TENG in atmosphere is unable to drive the temperature-humidity meter alone. **g** Thirty-two LED light bulbs (each with rated power of 0.75 W) are lit in complete darkness by the TENG in high vacuum, **h** Thirty-two LED light bulbs cannot be lit by the TENG in atmosphere, **i** two LED light bulbs are lit in complete darkness by the TENG in high vacuum; Scale bar, 4 cm

**Performance enhancement of TENG via coupling of surface and dielectric polarization.** Although the exact origin of triboelectrification is still under debate, it is known that materials with higher electron affinity, i.e., better capability of trapping electrons, can acquire more negative charges during the contact electrification process. Previous publications have reported the use of surface modification, the introduction of the electric double-layer effect, and the addition of a charge transport layer to enhance the triboelectric charges[29, 34]. Meanwhile, ferroelectric materials are known to have residual dielectric polarization after being exposed to an electrical field. The residual polarization, if properly coupled with the surface polarization on triboelectric materials, can be expected to enhance its ability to capture charges. Herein, a method of increasing triboelectric charge density by introducing built-in dielectric polarization using a ferroelectric material is proposed, and its working principle is schematically illustrated in Fig. 3a. The TENG consists of a top triboelectric Cu electrode (denoted as Cu-I), a PTFE film adhered to a ferroelectric material layer, and a back Cu electrode at the bottom (denoted as Cu-II). When Cu-I and PTFE are in contact, they will acquire net opposite charges on their surfaces. Once Cu-I is separated from the PTFE film, the induced potential

between two charged surfaces, or surface polarization, will result in a dielectric polarization inside the ferroelectric layer. Under short-circuit condition, the resultant surface polarization will also drive the positive charges on Cu-I to Cu-II, until the gap increases to the maximum and an equilibrium is reached. As the Cu-I approaches the PTFE, the decreased surface polarization will drive the positive charges on Cu-II back to Cu-I until Cu-I and PTFE are in contact again. Due to dielectric hysteresis, however, the polarization inside the ferroelectric material will not fully diminish, and the residual built-in dielectric polarization will act as a negative charge trap to enhance the PTFE's capability of capturing charges during contact electrification. In other words, the surface polarization from triboelectrification will induce built-in dielectric polarization inside the ferroelectric material, while the latter will enhance the former in subsequent contact electrification processes until an equilibrium is reached. This coupling of surface and dielectric polarization will greatly enhance the amount of triboelectric charges that can be generated during the operation of TENG.

To validate the proposed mechanism, one type of doped barium titanate material (namely BT, whose properties are characterized in Supplementary Fig. 4 and Supplementary

Note 4), was used as the ferroelectric material. The TENG has a circular contact area with a diameter of 1 cm, and its output performance is presented in Fig. 3b–d. The output charge density is enhanced from $142\,\mu C\,m^{-2}$ in air to $1003\,\mu C\,m^{-2}$ in high vacuum ($10^{-6}$ torr), and accordingly, the open circuit voltage increases from 26 V to 180 V and the peak of short-circuit current rises from 80 to $570\,mA\,m^{-2}$. The maximum output power density, with a load of 10 MΩ, is elevated from 1.1 to $50\,W\,m^{-2}$, which is over 45-fold enhancement (Fig. 3e,f). It should be noted that with only BT as the bottom triboelectric layer, the triboelectric charge density is lower than $15\,\mu C\,m^{-2}$ in atmosphere, while that with only PTFE is $120\,\mu C\,m^{-2}$ (Supplementary Fig. 5). By integrating PTFE with BT, on the other hand, the charge density is $142\,\mu C\,m^{-2}$ in atmosphere, around the theoretical limitation of air breakdown (Supplementary Fig. 6), and a record high of $1003\,\mu C\,m^{-2}$ in high vacuum without air breakdown. The results are clear evidence of the synergistic effect between surface and dielectric polarization. This mechanism opens a paradigm for TENG optimization via coupling of surface polarization from triboelectrification and hysteretic dielectric polarization from a ferroelectric material.

Selected milestones of TENG development are summarized in Fig. 3g. Initially, the triboelectric charge density of TENG based on Cu and PTFE is around $50\,\mu C\,m^{-2}$[23]. The value is improved to 134 and $219\,\mu C\,m^{-2}$ when liquid metal galinstan and gallium, instead of Cu, are used respectively, due to better contact intimacy[22]. Inspired by liquid metal, soft material and fragmental contacting structure (silicon rubber mixed with carbon materials, SR-C) TENGs reach $250\,\mu C\,m^{-2}$[28]. In this work, the charge density in high vacuum is enhanced to $660\,\mu C\,m^{-2}$ simply with a cushioned Cu electrode and a fragmental structure, and further elevated to $1003\,\mu C\,m^{-2}$ by integrating PTFE with ferroelectric BT. Compared with that of the conventional TENG as shown in Supplementary Fig. 2, the output triboelectric charge density has been boosted by 20-fold.

**Application of the TENG working in vacuum**. The elevated output performance of a TENG with a soft contact area of $5\,cm \times 5\,cm$ in high vacuum was demonstrated through driving various electronics together with an energy storage unit. Here a supercapacitor of 1.2 mF. Fig. 4a presents the circuit diagram of the self-powered system, where the TENG and the supercapacitor are connected by a full-wave rectifier. The voltage of the supercapacitor is monitored by a voltmeter. When switch K1 is on and switch K2 is off, the voltage of the supercapacitor is increased by 21.49 mV after being charged by the TENG in atmosphere for 10 s (Fig. 4b), and the equivalent galvanostatic current ($I_{eg}$) can be calculated as 2.5 μA (Supplementary Note 5). When the same TENG works in high vacuum ($P \sim 10^{-6}$ torr), the voltage of the supercapacitor increases by 80.36 mV in the same time period (means 4.8 V in 10 min), delivering an $I_{eg}$ of 9.6 μA, which is around 4 times of that in atmosphere.

Thanks to the elevated output performance in high vacuum, the TENG working at a low frequency of 2 Hz can not only drive an electronic watch sustainably, but also charge the super-capacitor simultaneously, indicating that the TENG generates more energy than the watch consumes (Fig. 4c, Supplementary Movie 1). At first, when the supercapacitor powers the electronic watch alone, its voltage declines with discharging, and the consumption current of the watch is calculated as 1.76 μA. When the TENG is operated, the voltage of the supercapacitor increases with a charging current of 7.2 μA while the electronic watch is powered simultaneously. Even though the TENG in atmosphere can drive both devices as well, the charging current of the supercapacitor is as low as 0.61 μA (Fig. 4d, Supplementary Movie 2).

For a device with higher power consumption, for instance a commercial humidity–temperature meter, only working in high vacuum allows the TENG to power the device and charge the supercapacitor simultaneously and sustainably (Fig. 4e, Supplementary Movie 3). The consumption current is 3.9 μA and the average charging current is 4.1 μA. When the TENG operates in air, the supercapacitor voltage declines, which means the TENG in atmosphere cannot drive the humidity–temperature meter alone (Fig. 4f, Supplementary Movie 4).

Furthermore, an LED light bulb array (4 groups in parallel × 8 bulbs in series, rated power 0.75 W × 32) can be lit by the TENG working in high vacuum (Fig. 4g, Supplementary Movie 5), but will not be lit by it working in atmosphere (Fig. 4h). Actually, the latter can only support two bulbs (Fig. 4i, Supplementary Movie 6). This brief but straightforward comparison demo shows the output power of the TENG is enhanced 16-fold in high vacuum.

**Discussion**

In this work, the triboelectric charge density of TENG is first improved to $660\,\mu C\,m^{-2}$ in vacuum where the limitation of air breakdown is eliminated, and further to $1003\,\mu C\,m^{-2}$ via coupling of surface and dielectric polarization. High vacuum environments cannot only guarantee better performance of TENGs, but also spare TENGs from performance degradation caused by the natural accumulation of dust and air moisture. The progress here sets a performance record for TENGs and establishes an optimization methodology for them. This work also provides an insight into the restricting factors on performance of TENGs, making it necessary to re-estimate the upper limit of TENGs to be much higher than previously expected.

The surface charge density of TENG is simultaneously limited by the triboelectrification charge density, air breakdown and dielectric breakdown, described formally as follows:

$$\sigma_{TENG} = \min(\sigma_{triboelectrification},\, \sigma_{air\_breakdown},\, \sigma_{dielectric\_breakdown})$$

$$(3)$$

Without the concern of air breakdown (Supplementary Fig. 1 and Supplementary Note 1), dielectric breakdown may become the next bottleneck of TENG. Concerning the PTFE film, its $\sigma_{dielectric\_breakdown}$ is calculated to be around $1115\,\mu C\,m^{-2}$ (Supplementary Fig. 7 and Supplementary Note 6), which is not far from our result of $1003\,\mu C\,m^{-2}$. Future work will involve the optimization of dielectric material to further explore the limitations of triboelectrification.

Without loss of generality, the paradigm to enhance the triboelectric charge density of TENG in this work can be applied to other technologies involving contact electrification. Furthermore, it will benefit the long-lasting debate over the underlying mechanism of triboelectrification and its kinetics, which calls for more experimental evidence to test existing hypotheses, such as electron transfer by tunneling effect, mass transport, and even a hybrid of both[35]. In high vacuum, interfering factors with triboelectrification such as dust and moisture can be reduced or even eliminated, and a much higher amount of transferred charge can be detected, both of which will favor the evidence seeking process.

In practice, our study points to an effective approach for enhancing the output power of TENGs, which greatly improves the prospect of large-scale blue energy using nanogenerators networks[36]. Considering that existing TENGs proposed for harvesting water wave energy are already enclosed and sealed with waterproof containers[37, 38], it is rather straightforward and cost-effective to make them airtight and maintain an internal vacuum. The resulted boost in electrical output can reduce the

required spread of blue-energy nets, and thus minimize the environmental impacts while meeting energy needs especially in extreme weather conditions.

## Methods

**Fabrication of the TENG**. With a conventional TENG with hard contact used as a reference, a Cu film (Cu-I) was adhered to an acrylic substrate, acting as an electrode and a triboelectric material simultaneously, while the other triboelectric material, a PTFE film, was adhered to another Cu film (Cu-II) and then onto an acrylic substrate. The Cu-I, Cu-II and PTFE film were all of the same size, $3 \times 3\,cm^2$. To construct the TENG with a cushioned Cu electrode and the fragmental structure, a piece of foam was placed between Cu-I and its corresponding acrylic substrate, and the PTFE and Cu-II films were reduced to a smaller size with a circular diameter of 1 cm. For the TENG with ferroelectric material, a piece of BT ceramic was placed between the PTFE film and Cu-II, with all others parameters kept the same (Supplementary Fig. 8). The thickness of the PTFE film was 200 µm, and three layers of PTFE were stacked in the TENG for studying the thickness effect. The synthesis process of BT ceramic samples is depicted in Supplementary Note 7. The TENG used for application demonstration in Fig. 4 had a cushioned Cu electrode and did not involve ferroelectric material. Its Cu-I, Cu-II and PTFE film all had the same size of $5 \times 5\,cm^2$.

**COMSOL simulation**. The 2D potential distribution between two oppositely charged surfaces of TENG, as plotted in Fig. 1d, was numerically calculated using the commercial software COMSOL. The width of the device was set to be 1 cm and the thickness of both Cu and PTFE film was 200 µm. The gap distance between the Cu and PTFE film was 1 cm, and the surface charge density was $50\,\mu C\,m^{-2}$ in Fig. 1d.

**Characterization**. The vacuum environment for the TENG testing was obtained using an e-beam evaporator (Key). The contact-separation process of the TENG device was driven by the movement of a target shutter inside the vacuum chamber of the evaporator, and the external force was applied to an external knob which was connected to the target shutter. A programmable electrometer (Keithley model 6514) was adopted to test the open-circuit voltage, short-circuit current, and transferred charge density. A potentiostat (Princeton Application Research) was utilized to test the capacitance of the capacitor and the charging/discharging curves of the self-charging power system. The characterization of BT ceramic is depicted in Supplementary Note 8.

**Data availability**. The data supporting the findings of this study are available from the corresponding author on request.

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

## Acknowledgements

Research was supported by the Hightower Chair foundation, and the "thousands talents" program for pioneer researcher and his innovation team, China, the National Key R & D Project from Minister of Science and Technology (2016YFA0202704), National Natural

Science Foundation of China (Grant No. 51432005, 5151101243, 51561145021). Patents have been filed based on the research results presented in this manuscript.

## Author contributions

J.W., C.W., Y.D., and Z.L.W. conceived the idea, analyzed the data, and wrote the paper. J.W. and C.W. designed the structures of the triboelectricnanogenerators. Y.D. and Z.Z. prepared and characterized the ferroelectric material. A.W. and T.Z. helped with the experiments. All the authors discussed the results and commented on the manuscript.

## Additional information

**Competing interests:** The authors declare no competing financial interests.

