## [Peer Review File · Nature Communications]

Reviewers' comments:

Reviewer #1 (Remarks to the Author):

The manuscript present a new approach for applying TENGs in Vacuum which is very interesting, Here are few questions to be addressed,

1. in line 184-186, author mentioned that BT and PTFE charge density and when they merge together, the charge density is higher than add both of them simply, does the author try other materials and how about their difference? What is the principle behind?
2. in line 142, PTFE thickness change from 200-600um, how about Cu film thickness? Does it affect the results or not? can it agree with the COMSOL simulation results?
3. authors cited too many publications from same group, there are many other publications from other group in same field need to be mentioned.

Reviewer #2 (Remarks to the Author):

This manuscript reported a new method to increase the TENG performance using vacuum. The device showed in the manuscript set a new performance record and provides guidelines for future device design and fabrication. This is an milestone work in the field of nanogenerators and will be of interests to many researchers in the field. The manuscript is well written and work is clearly presented. Some spaces between words are missing in the manuscript, which can be easily fixed. This reviewer support the publication of this manuscript.

Reviewer #3 (Remarks to the Author):

Manuscript Number: NCOMMS-17-05541-T

Achieving ultrahigh triboelectric charge density via coupling of surface and dielectric polarization in vacuum for high efficient energy harvesting

In this manuscript a method for increasing the triboelectric charge density by coupling surface polarization from triboelectrification and hysteretic dielectric polarization from ferroelectric material in vacuum ($P \sim 10^{-6}$ torr) is provided. Without the constraint of air breakdown, a record- triboelectric charge density of $1003 \mu\text{Cm}^{-2}$, which is close to the limit of dielectric breakdown, is attained. I read through the manuscript based on the idea and presented data at this submission, and found that there is novelty over the previous studies.

I think that the novelty of this manuscript is sound enough and I recommend minor revision for this manuscript. Some questions should be addressed.

1. Authors have been discussing about a TENG device in the manuscript, but there is no sign of that device (digital image) in the whole manuscript.
2. In this manuscript authors are mainly discussing about the performance of TENG in vacuum atmosphere, but why exactly $P \sim 10^{-6}$ torr is applied for the device characterization and why not above or below this value of vacuum?
3. Authors should mention in the experimental section how they applied external force to the TENG device, when it was kept in vacuum condition.
4. Several supplementary video are provided by the authors, but in that no video shows the working of the proposed TENG device. Only LED bulb glowing and capacitor charging is shown.
5. The relative references about triboelectric nanogenerator need to be cited in this paper. ACS Appl. Mater. Interfaces, 2016, 8 (15), pp 9692–9699, J. Mater. Chem. C, 2017,5, 1488-1493.
6. Authors should check the typographical errors.
Some examples
1) Line no: 111 "theVgapmust"
2) Line no: 135 "operatedin vacuum"
3) Line no: 125 "143 $\mu\text{C m}^{-2}$ "
4) Line no: 178 " whosepropertiesare"
5) Line no: 207 "atmospherefor"
6) Line no: 218 "simultaneously.Even"

Point-by-point responses to the reviewers' comments

We sincerely thank the reviewers for their careful and thorough review, which are indeed very helpful to make the paper more solid and smooth. We have revised our manuscript very carefully in the light of their suggestions and comments.

The following responses have been prepared to address all of the reviewers' comments in a point-by-point fashion. **(Comments in black, responses in Red):**

Reviewer #1 (Remarks to the Author):

The manuscript present a new approach for applying TENGs in Vacuum which is very interesting, Here are few questions to be addressed,

Response: Thank the reviewer for the positive feedback and we will try our best to address the questions raised.

1. in line 184-186, author mentioned that BT and PTFE charge density and when they merge together, the charge density is higher than add both of them simply, does the author try other materials and how about their difference? What is the principle behind?

Response: Yes, we have also tried the combination of Silicone and BT. When contacting with copper electrode in air, the charge density of Silicone is around $150 \mu\text{C m}^{-2}$, and the value of combination of Silicone and BT is up to $220 \mu\text{C m}^{-2}$, which is greater than the sum of charge density value of Silicone only ($150 \mu\text{C m}^{-2}$) and the one of BT only ($15 \mu\text{C m}^{-2}$). This is another evidence of synergistic effect.

As proposed in the manuscript, the synergistic effect may be attributed to the coupling of surface polarization from triboelectrification and hysteretic dielectric polarization from ferroelectric material. The surface polarization from triboelectrification will induce built-in dielectric polarization inside the ferroelectric material, while the latter will enhance the former in subsequent contact electrification processes until an equilibrium is reached, as illustrated by Fig. 3a. This proposed mechanism is also evidenced by the gradual increase of surface charge density during the initial operation cycles of the PTFE/BT based TENG in vacuum, from $90 \mu\text{C m}^{-2}$ at initial period to $1003 \mu\text{C m}^{-2}$ after around 30 min operation.

Figure 1. The charge density of PTFE/BT based TENG increasing with working time in vacuum.

2. in line 142, PTFE thickness change from 200-600um, how about Cu film thickness? Does it affect the results or not? can it agree with the COMSOL simulation results?

Response: In our case, all Cu films have the thickness of 500 um. However, the Cu thickness would not affect either the charge generation process or charge transport process given that (a) the contact electrification takes place at the material surfaces and (b) Cu is a very good conductor, and thus it should not affect the results.

In the COMSOL model, identical results were obtained when the Cu film thickness was adjusted alone. The change in Cu film thickness has no influence on the potential distribution, which is the driving force of charge redistribution during TENG operation.

3. authors cited too many publications from same group, there are many other publications from other group in same field need to be mentioned.

Response: Thank the reviewer for the suggestion. We have added several relevant references from other groups.

The added references are listed as follow:

7 Chandrasekhar, A., Alluri, N. R., Vivekananthan, V., Purusothaman, Y. & Kim, S.-J. A sustainable freestanding biomechanical energy harvesting smart backpack as a portable-wearable power source. *Journal Of Materials Chemistry C* **5**, 1488-1493, doi:10.1039/c6tc05282g (2017).

8 Zhu, M. *et al.* 3D spacer fabric based multifunctional triboelectric nanogenerator with great feasibility for mechanized large-scale production. *Nano Energy* **27**, 439-446, doi:10.1016/j.nanoen.2016.07.016 (2016).

16 Chandrasekhar, A., Alluri, N. R., Saravanakumar, B., Selvarajan, S. & Kim, S.-J. Human Interactive Triboelectric Nanogenerator as a Self-Powered Smart Seat. *Acs Applied Materials & Interfaces* **8**, 9692-9699, doi:10.1021/acsami.6b00548 (2016).

17 Zhang, X., Zheng, Y., Wang, D., Rahman, Z. U. & Zhou, F. Liquid-solid contact triboelectrification and its use in self-powered nanosensor for detecting organics in water. *Nano Energy* **30**, 321-329, doi:10.1016/j.nanoen.2016.10.025 (2016).

18 Rodrigues, C. R. S., Alves, C. A. S., Puga, J., Pereira, A. M. & Ventura, J. O. Triboelectric driven turbine to generate electricity from the motion of water. *Nano Energy* **30**, 379-386, doi:10.1016/j.nanoen.2016.09.038 (2016).

Reviewer #2 (Remarks to the Author):

This manuscript reported a new method to increase the TENG performance using vacuum. The device showed in the manuscript set a new performance record and provides guidelines for future device design and fabrication. This is an milestone work in the field of nanogenerators and will be of interests to many researchers in the field. The manuscript is well written and work is clearly presented. Some spaces between words are missing in the manuscript, which can be easily fixed. This reviewer support the publication of this manuscript.

Response: Thank the reviewer for the positive comments. We have gone through the manuscript carefully and fixed all the typos.

Reviewer #3 (Remarks to the Author):

Manuscript Number: NCOMMS-17-05541-T

Achieving ultrahigh triboelectric charge density via coupling of surface and dielectric polarization in vacuum for high efficient energy harvesting

In this manuscript a method for increasing the triboelectric charge density by coupling surface polarization from triboelectrification and hysteretic dielectric polarization from ferroelectric material in vacuum ($P \sim 10^{-6}$ torr) is provided. Without the constraint of air breakdown, a record- triboelectric charge density of $1003 \mu\text{Cm}^{-2}$, which is close to the limit of dielectric breakdown, is attained. I read through the manuscript based on the idea and presented data at this submission, and found that there is novelty over the previous studies.

I think that the novelty of this manuscript is sound enough and I recommend minor revision for this manuscript. Some questions should be addressed.

Response: Thank the reviewer for the positive feedback and we will try our best to address the questions below.

1. *Authors have been discussing about a TENG device in the manuscript, but there is no sign of that device (digital image) in the whole manuscript.*

Response: Thank the reviewer for pointing this out. We have added an image of the real TENG device in Supplementary Fig 8.

Supplementary Fig.8 Image of the TENG with the coupling of surface and dielectric polarization.

2. *In this manuscript authors are mainly discussing about the performance of TENG in vacuum atmosphere, but why exactly $P \sim 10^{-6}$ torr is applied for the device characterization and why not above or below this value of vacuum?*

Response: The vacuum environment for the TENG testing was obtained using an e-beam evaporator (Key), which is an old system that does not have the capability of maintaining an arbitrary pressure. Its chamber pressure at equilibrium was around 10^{-6} torr and thus it was applied for the device characterization. At the 10^{-6} torr vacuum, the TENG operation range is outside of the region where Paschen's law holds valid or avalanche breakdown is possible, and thus we believe that this does not affect the soundness and completeness of this work.

We totally agree that the study of performance at different values of vacuum is meaningful and can help decide the optimal value for practical applications, and we have planned to study this with upgraded equipment in the future.

3. *Authors should mention in the experimental section how they applied external force to the TENG device, when it was kept in vacuum condition.*

Response: Thank the reviewer for the suggestion. The evaporator used has target shutters inside the vacuum chamber and these shutters are connected to external knobs through a set of gears and beams to realize airtightness and mechanical transmission simultaneously. During experiment, we applied external force to control the rotation of the knob at atmosphere and thus the movement of the target shutter in vacuum, which drove the contact-separation process of the TENG device.

We have added this in the Methods section.

"... an e-beam evaporator (Key). The contact-separation process of the TENG device was driven by the movement of a target shutter inside the vacuum chamber of the evaporator, and the external force was applied to an external knob which was connected to the target shutter. A programmable electrometer ..."

4. *Several supplementary videos are provided by the authors, but in that no video shows the working of the proposed TENG device. Only LED bulb glowing and capacitor charging is shown.*

Response: Since the device was inside the fully enclosed vacuum chamber and there was no camera inside it, the working of device in vacuum was not recorded in the videos. For all the measurements and demonstrations, the electrodes of the TENG device were connected to external cables outside the vacuum chamber through built-in conductive channels. To better present the results, only applied loads and test screens, but not the big closed chamber, were included in the videos.

5. *The relative references about triboelectric nanogenerator need to be cited in this paper. ACS Appl. Mater. Interfaces, 2016, 8 (15), pp 9692–9699, J. Mater. Chem. C, 2017,5, 1488-1493.*

Response: Thank the reviewer for the suggestion. Some additional references including the two ones mentioned have been added.

6. *Authors should check the typographical errors.*

Some examples 1) Line no: 111 “theVgapmust”

2) Line no: 135 “operatedin vacuum”

3) Line no: 125 “143 μ C m-2”

4) Line no: 178 “ whosepropertiesare”

5) Line no: 207 “atmospherefor”

6) Line no: 218 “simultaneously.Even”

Response: Thank the reviewer for pointing them out. We have corrected the errors accordingly.

REVIEWERS' COMMENTS:

Reviewer #1 (Remarks to the Author):

The author answered my questions point to point, the results is reasonable and quality is high, ready to be accept for publish.

Reviewer #3 (Remarks to the Author):

Authors have addressed all the questions in detail.

Respond to reviewers' comments:

Reviewer #1 (Remarks to the Author):

The author answered my questions point to point, the results is reasonable and quality is high, ready to be accept for publish.

Response: Thank you very much for your comments.

Reviewer #3 (Remarks to the Author):

Authors have addressed all the questions in detail.

Response: Thank you very much for your comments.